# Vocalization with semi-occluded airways is favorable for optimizing sound production

Ingo R. Titze[1,2,3]*, Anil Palaparthi[1,2,3], Karin Cox[3], Amanda Stark[1], Lynn Maxfield[1], Brian Manternach[1]

**1** National Center for Voice and Speech University of Utah, Salt Lake City, Utah, United States of America,
**2** Department of Biomedical Engineering, University of Utah, Salt Lake City, Utah, United States of America,
**3** National Center for Voice and Speech.Org, Salt Lake City, Utah, United States of America

* ingo.titze@utah.edu

## Abstract

Vocalization in mammals, birds, reptiles, and amphibians occurs with airways that have wide openings to free-space for efficient sound radiation, but sound is also produced with occluded or semi-occluded airways that have small openings to free-space. It is hypothesized that pressures produced inside the airway with semi-occluded vocalizations have an overall widening effect on the airway. This overall widening then provides more opportunity to produce wide-narrow contrasts along the airway for variation in sound quality and loudness. For human vocalization described here, special emphasis is placed on the epilaryngeal airway, which can be adjusted for optimal aerodynamic power transfer and for optimal acoustic source-airway interaction. The methodology is three-fold, (1) geometric measurement of airway dimensions from CT scans, (2) aerodynamic and acoustic impedance calculation of the airways, and (3) simulation of acoustic signals with a self-oscillating computational model of the sound source and wave propagation.

## Author summary

Humans and animals communicate vocally. For mammals, birds, and reptiles that breathe with lungs, a source of sound is embedded in the airways, somewhere between the lungs and the lips or beak. When the mouth or beak is open, sound radiates into free-space to be received by listeners. The wider the opening, the more sound is released from the airway. However, for centuries those who teach efficient voice production have encouraged their clients to practice with a nearly closed mouth, therefore partially occluding the airway. In the animal world, doves, frogs, and gerbils vocalize with completely closed airways, allowing the sound to radiate from vibrating skin surfaces instead of airway openings. It is shown here that by keeping most of the sound inside the body and widening the throat by vocalizing into a narrow tube at the mouth, the sound source can become more efficient. By controlling the airspace in the larynx directly above the vocal cords, an area known as the epilaryngeal airway, this efficiency can be maintained when the mouth is opened after practice. A large variety of sound qualities and a louder voice are thereby facilitated without a significant effect on vowel production.

**Data Availability Statement:** All relevant data are within the manuscript.

**Funding:** I.T. was awarded funding through the NIDCD R01 DC013573. The funders had no role in

study design, data collection and analysis, decision to publish, or preparation of the manuscript.

**Competing interests:** The authors have declared that no competing interests exist.

## Introduction

Vocalization in mammals, birds, and reptiles involves one or more sound sources that self-sustain oscillation in an airway. The oscillation requires interaction between a moving fluid and a soft wall structure and is often facilitated by acoustic waves propagating in the airway. Morphological requirements for sound production are generally secondary to requirements for respiration and ingestion, but if sound production were the dominant requirement, it is unclear what the ideal airway geometry would be, especially with regard to length, shape, and the size of openings for sound release. Nature provides multiple options, but they trend in different directions. On the one hand, doves, gerbils, and frogs widen the airway and occlude (or partially occlude) the mouth or beak, radiating sound from puffed-up neck surfaces [1–3]. On the other hand, long-range calling by other birds and mammals requires extreme mouth or beak openings [4–12]. In terms of the optimal location of the sound source within the airway, a recent investigation by Riede et al. [13] showed that a sound source deep within the airway (closer to the lungs, with most of the airway in front of it) produces sound more efficiently than a source in close proximity to the mouth or beak (with most of the airway behind it). Acoustic interaction between the source and the airway plays a major role. The airway stores acoustic energy, which is fed back to the source. With the right phase relations between acoustic pressures and vocal fold movement, the source output can be strengthened.

Musical wind instruments exhibit the same interaction phenomena between the source and the resonator (e.g., Adachi and Sato, 1996, for trumpet playing) [14], but here the biological airway is augmented with extra tubes or horns. The main difference between vocalization and playing wind instruments is the nature and location of the sound source (a reed or lips instead of vocal folds) and the artificial extension of the airway. In vocalization, it is now beginning to be understood that an extension of the vocal tract with a tube, a straw, a megaphone, or other device can be beneficial in "training" the sound source for less-effortful production with greater efficiency [15]. In particular, a narrow tube extended from the mouth produces a steady airway pressure that (as will be shown here) can lead to an optimal airway configuration and optimal posturing and shaping of the vocal folds for vibration. This has led to exercises conducted with a semi-occluded vocal tract (SOVT) [16–18], perhaps capitalizing on the source-airway interaction routinely experienced by doves and frogs, or the variable interaction experienced by brass instrument players who mute the instrument at the bell [12].

Quantification of the interaction between the source and the airway in human vocalization began with the mathematical model of Flanagan and Landgraf [19]. They showed that a simple one-mass vocal fold model could self-sustain oscillation due to fluid-structure interaction, but the oscillation was highly dependent on the acoustic properties of the airways above and below the vocal folds. Ishizaka and Flanagan [20] then showed that an acoustically compliant airway above the sound source squelched the oscillation, while an acoustically inertive airway supported the oscillation.

It is now understood that nonlinear relationships exist between the opening and closing movement of an oscillating valve (vocal folds, lips, reeds), the pressures above and below the valve, and the airflow through the valve [21,22]. Movement of the oscillating valve structure can be very smooth (nearly sinusoidal) while movement of the air through the valve can be rapidly changing (square, pulse, or saw-tooth-like). The peak of the airflow is often delayed relative to the peak of the valve movement [23–26]. It has further been shown that air inertance of the vocal tract can lower the oscillation threshold pressure [22,27].

Interaction between the airway and glottal airflow (acoustic-aerodynamic interaction) and interaction between acoustic pressures and vocal fold movement (mechano-acoustic interaction) increase when the impedance of the airway is on the same order of magnitude as the

impedance of the source, which is dominated by the glottal resistance. For a wide-open airway at the mouth and a typical range of speech fundamental frequencies, the acoustic airway impedance is generally much lower than the source impedance, resulting in less interaction [28]. An exception occurs at an airway resonance, where the impedance is high. For all frequencies below the first resonance frequency of the airway, if a portion of the airway is narrowed (or semi-occluded), the impedance can rise to a level closer to that of the source (referred to as impedance matching), increasing the interaction as well as the power transfer. This principle is also realized in sound reception by the auditory system. The narrow ear canal matches the impedance of free space to the impedance of the tympanic membrane [29]. In addition to narrowing a portion of the airway, it is also possible to alter the impedance by changing the length of the airways, but this strategy is more difficult in biological systems because airway lengths are relatively constant and conservative. In birds, however, airway lengthening has evolved dramatically over long time periods [13]. The tracheal airway, coiled inside the thorax, can exceed the length of the entire body of the bird. In humans, artificial lengthening is possible with a tube or straw at the mouth.

Laukkanen et al. [30] collected MRI images of one female phonating an [a:] vowel, then phonating into a plastic drinking straw between the lips with no tongue change, and then again phonating an [a:]. They found that during and after phonation into the straw, the mid-sagittal area of the vocal tract increased and there was greater velar closure. For this subject, the ratio of the cross-sectional area of the lower pharynx to that of the epilaryngeal airway increased both during straw phonation (27%) and after straw phonation (20%). Guzman et al. [31] and Guzman et al. [32] have followed up with similar studies using CT imaging. They observed that pharyngeal widening and larynx lowering occurred during semi-occluded vocal tract exercises (SOVTE). Their results will be analyzed further in this paper.

For phonetic and voice quality contrast in speech and singing, it is necessary to widen portions of the airway while simultaneously narrowing other portions. This selective widening and narrowing is generally accomplished with articulatory gestures. A greater wide-narrow contrast can be achieved by first widening the airway uniformly, a technique to be discussed in this paper. Selective narrowing can then be superimposed. Unfortunately, active (muscular) widening of the airway walls is not a frequent need in humans, other than for deep breathing during or after strenuous exercise, or for singing long phrases that require a large expiratory lung volume. Without these activities, passive widening is possible by introducing a positive pressure in the upper airway with an airway semi-occlusion, as will be shown in this study.

With regard to airway narrowing, a region above the vocal folds, the epilaryngeal airway (labeled Epi-Larynx Tube in Fig 1), is naturally narrow and can serve to match the vocal fold airflow impedance to the impedance of the airways, like the ear canal to tympanic membrane motion, or the mouth-piece of a trumpet to the vibrating lips. The epilaryngeal airway consists of the ventricle, the ventricular space between the false folds, and the laryngeal vestibule (differentiated in the coronal portion of the sketch in Fig 2 in green color). Pharyngeal widening or narrowing (blue portion of Fig 2, rotated 90 degrees to a sagittal view) can then produce further airway shape contrast.

Vocal exercises have been devised to alter the shape of the airways. The exercises are known as semi-occluded vocal tract exercises (SOVTEs). Examples are lip tills, tongue trills, phonation into a straw or tubes, and the use of voiced fricatives or nasal continuants in speech [16,18,30,34–38]. Decades of anecdotal reports have revealed that SOVTEs result in easier phonation, reduced vocal effort, and improved laryngeal health. However, only recently have SOVTEs begun to be validated in terms of their scientific underpinnings [17,18,30,32,38–44]. The exact physiological change, or series of changes that take place, is yet unknown. Descriptive studies of airway shape changes have utilized endoscopy, MRI, and CT imaging.

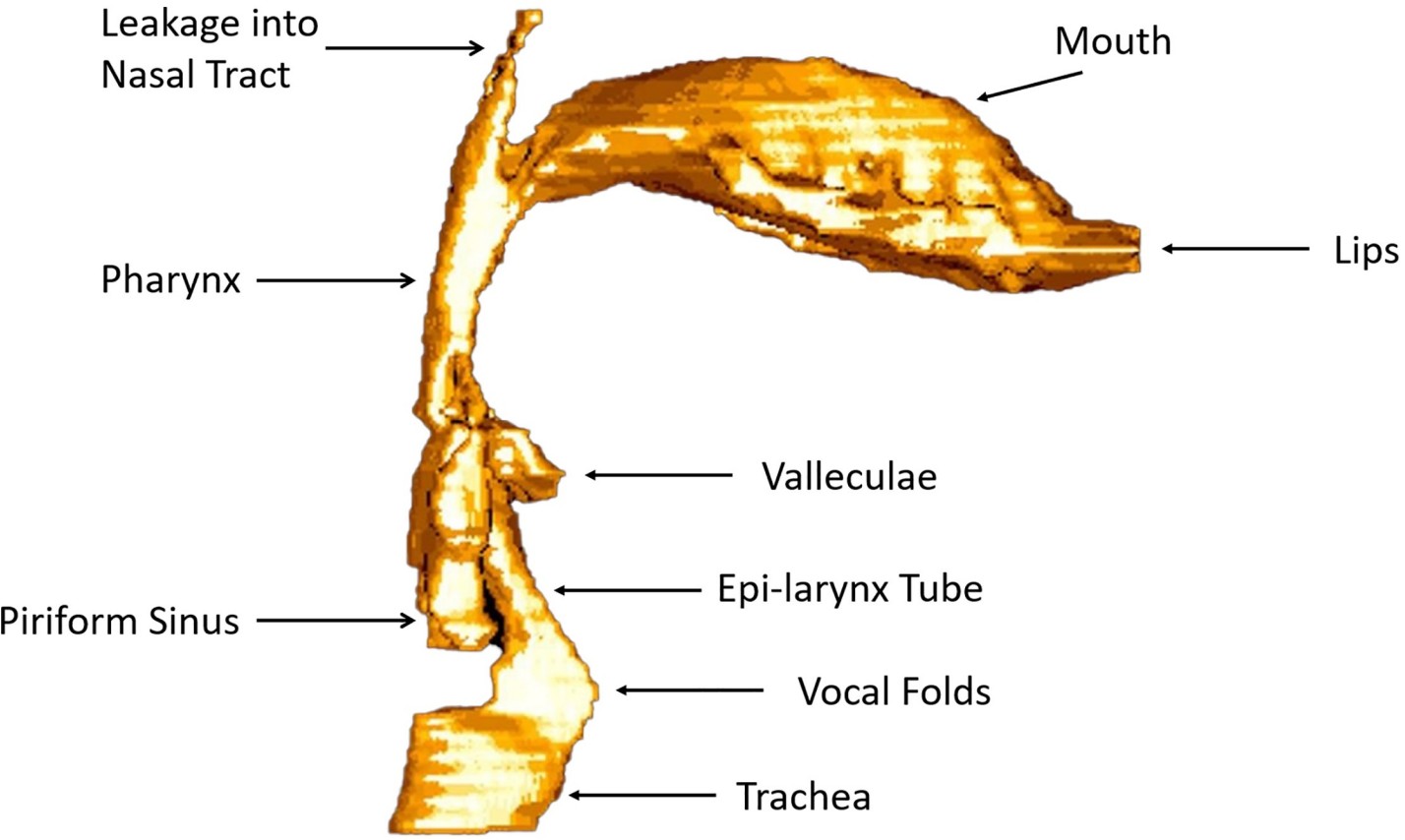

**Fig 1. Mid-sagittal MRI scan of a male human airway in an /ɑ/ vowel configuration.** (after Story B, et al.) [33].

The purpose of this study was to address the following hypotheses regarding phonation with semi-occluded airways: (1) Oral pressures widen all the airways passively, (2) If the epilaryngeal airway is selectively prevented from passive expansion, higher harmonic energy can be maximized with source-airway impedance matching, (3) The upper portion of the vocal folds are spread apart so that the medial surfaces of the vocal folds can be driven more effectively by acoustic airway pressures. Most of the benefits for phonation appear to be obtained passively with a steady pressure against the airway walls, but selective muscle activation is likely needed to maintain the idealized configuration when the semi-occlusion is removed.

## Results

### Airway expansions with an oral semi-occlusion

A digital airway was constructed with short circular or elliptical sections. Except for the glottal sections, all section lengths were chosen to be 0.3968 cm according to previous simulations [45–47]. This section length has become somewhat of a standard for discretizing the airway at a 44.1 kHz sampling frequency for wave propagation. With this section length, there were 36 circular sections for the trachea (14.3 cm total length), 2 for a transition into the glottis (0.79 cm total length), and 44 for the supraglottal tract (17.5 cm total length). Five glottal sections were shorter and elliptical, as described in the Methods section, with a total glottal length of 0.8 cm. A flow-resistant tube was added at the lips, the length and diameter of which was variable to produce a desired flow resistance. A two-dimensional sketch of the airway diameters

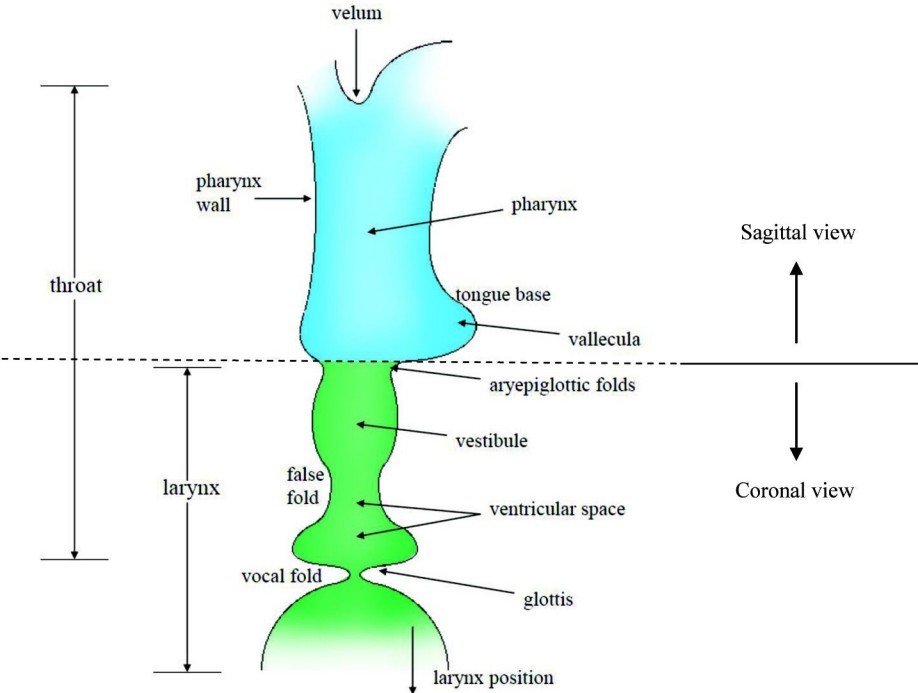

**Fig 2. Part coronal and part mid-sagittal sketch of the lower supraglottal airway (from Titze I and Verdolini-Abbott K, 2012) [15].**

from the entry to the trachea to the end of a lip tube is shown in Fig 3. A nasal airway, shown above the zero diameter line, was kept closed to airflow and acoustic waves so that the semi-occlusion was singular at the mouth. The model was used to produce airway wall expansion and sound computationally with self-sustained vocal fold oscillation and wave propagation. Details of the wall properties and the fluid solution are given in the Methods section.

The lung pressure was set to 2.5 kPa (25.5 cm $H_2O$) and a fundamental frequency of 300 Hz was targeted with elastic properties of the vocal folds, in the middle of a typical human singing range (80–500 Hz for males and 100–800 Hz for female) [48]. This is a higher fundamental frequency and a higher lung pressure than is typical for speech, but the elevations are a requirement for phonation into a flow-resistant tube [15]. The fundamental frequency was obtained

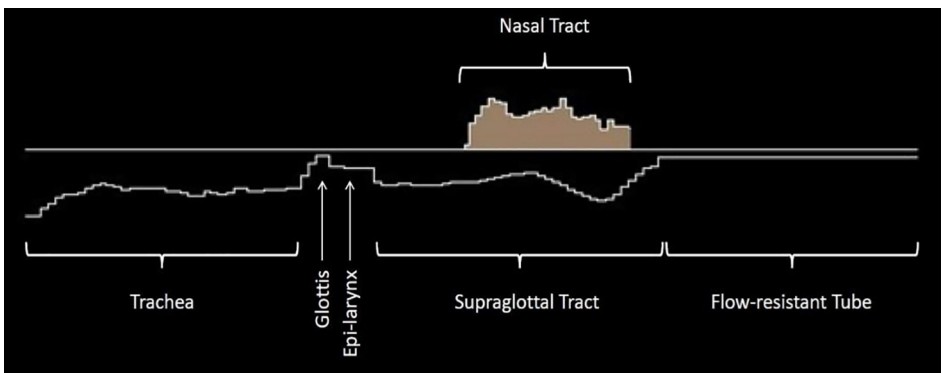

**Fig 3. Airway model with (left to right) a trachea, glottis, epi-larynx, supraglottal tract, a nasal tract, and flow-resistant tube (3 mm diameter and 10 cm length) as a semi-occlusion.**

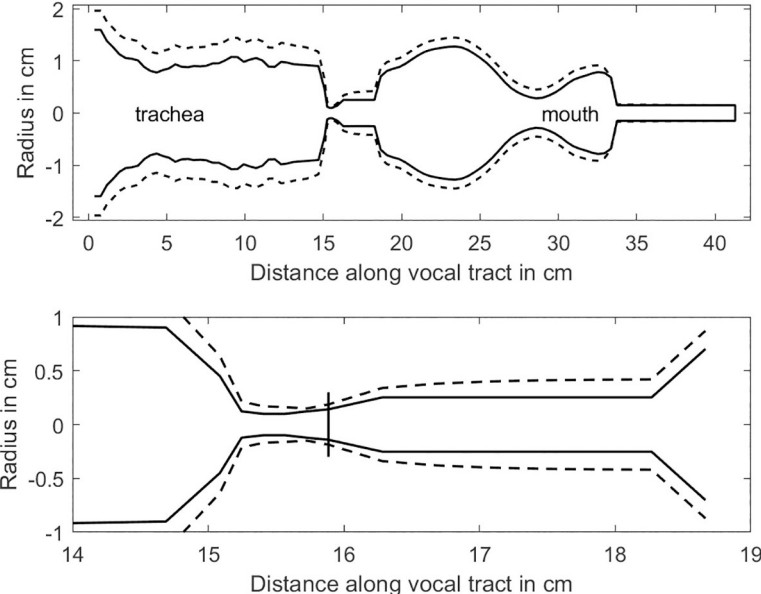

**Fig 4. Expansion of the airway in a simulation model with a 3.0 mm diameter flow-resistant tube as an oral semi-occlusion and a lung pressure of 2.5 kPa.** (top) entire airway, and (bottom) glottal and epilaryngeal airways expanded.

by adjusting the Young's modulus and shear modulus of the surface tissue and the mass per unit area, as described in the Methods section. Results of the airway expansion are shown in Fig 4. Solid lines show airway radii before lung pressure was applied (pre-pressure) and dashed lines indicate steady-state radii after lung pressure was applied (post-pressure) with the 3.0 mm diameter tube semi-occlusion. The top diagram shows the full length of the airway and the bottom diagram shows an expanded version of the glottal and epilaryngeal airways. Note that there is a uniform expansion of all portions of the airway. The expansion is greater in the trachea than in the supraglottal airway because pressures are generally higher in the trachea. Of particular interest is the expansion of the epilaryngeal airway. At exit (18.4 cm on the axis, bottom figure), the diameter has nearly doubled. Considerable widening also occurs in the pharynx and in the mouth. Without the oral semi-occlusions, only a tracheal expansion would occur because the mean supraglottal pressure is zero when the mouth is open.

Of equal importance is the expansion of the glottis. This is shown in Fig 5 for four different lip tube diameters (3,4,5,6 mm). For a linearly convergent glottis (entry wider than exit), the shape becomes more curved (bulging toward the midline around sections 3 and 4). This is highly advantageous for self-sustained oscillation because it offers both glottal convergence and divergence. Optimal driving pressures are obtained on the glottal surfaces when convergence and divergence alternate in vibration [49]. It has been shown that the medial surface shape of the vocal folds can dramatically affect the phonation threshold pressure [50]. If the pre-phonatory shape is not nearly rectangular or slightly bulging toward the midline, the alternating shapes in vibration are not likely to occur and self-sustained oscillation is inhibited. Note that the wider oral tubes (5 mm and 6 mm) produced only a modest glottal expansion, whereas the 3 mm tube dramatically separated the vocal folds at glottal entry and exit. This is a direct result of greater oral pressure produced with the higher tube resistance.

The airway expansions shown above were for a single lung pressure, 2.5 kPa. In the model, the change in radius is proportional to lung pressure because the wall stiffnesses remain constant with displacement (see Methods). In human airways, the expansions will likely diminish

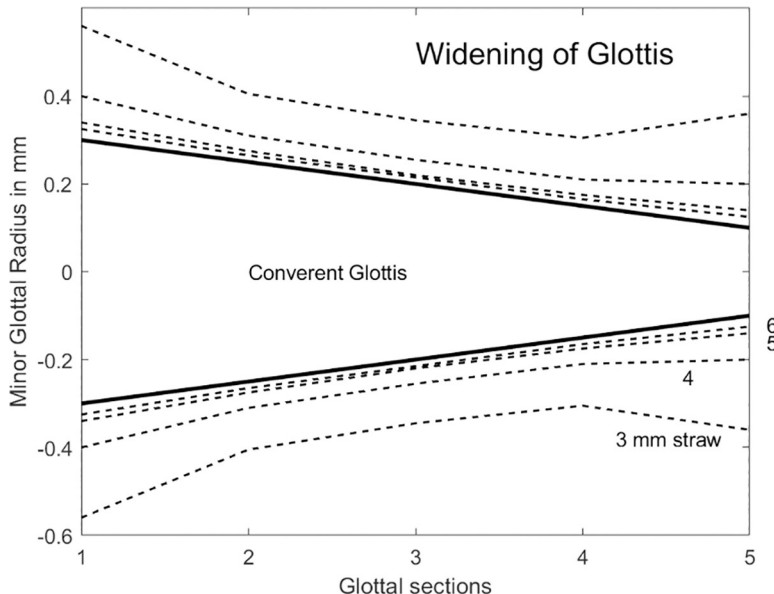

**Fig 5. Expansion of the minor diameters of the elliptical glottal sections with 2.5 kPa lung pressure and 3.0, 4.0, 5.0, and 6.0 mm tube diameters at the mouth.** The vertical to horizontal aspect ratio is 6.67 (5 sections have 8 mm length).

with high lung pressures due to nonlinear wall stiffness, but this nonlinearity has not been quantified.

To relate the above computations to measurements on CT scans described in the Methods section, Table 1 shows comparisons of several important airway expansions. In groups of three columns, we show measurements from Guzman's five male and five female human subjects before and during an oral semi-occlusion with a tube (Pre Guz, Dur Guz, and % change), from the simulation model (Pre Model, Dur Model, and % change) with male dimensions, and from a classical MRI study on a single male subject (recall Fig 1) [33]who produced an /ɑ/ vowel and an /m/ nasal consonant (/ɑ/ S.S., /m/ S.S., and % change). The consonant /m/ is considered a mild semi-occlusion of the airways, for which the mouth is fully occluded and the nostrils are the semi-occlusion. Nasals are used often in SOVT therapy.

**Table 1. Anatomical distances (mm) and cross-sectional areas (mm$^2$) obtained from CT scans of 10 subjects by Guzman et al., (Guz), the airway simulation model (Model), and MRI scans from a single male subject (S.S.) pre and during semi-occlusion (SOVT).**

| Measurement (refer to Fig 3 for landmarks) | Pre Guz | Dur Guz | % Change | Pre Model | Dur. Model | % Change | /ɑ/ S.S. | /m/ S.S. | % Change |
|---|---|---|---|---|---|---|---|---|---|
| Vertical pharyngeal tract length (FE) | 62 ± 16 | 72 ± 27 | +16% | 79.4 | 79.4 | --- | --- | ---- | ---- |
| Epilarynx Tract length (GE) | 14 | 16 | +14% | 23.8 | 23.8 | --- | ---- | ---- | ---- |
| Pharyngeal entry area (AD) | 47 ± 20 | 62 ± 24 | +32% | 155 | 239 | +54% | 105 | 218 | +107% |
| Epilarynx exit area (BC) | 14 ± 3.4 | 15 ± 9 | +7% | 20 | 55.3 | +177% | 33 | 58 | +76% |
| Epilarynx Entry area | --- | --- | --- | 20 | 36.3 | +82% | 45 | 57 | +27% |
| Glottal exit area | --- | --- | --- | 6.3 | 11.0 | +75% | ---- | ---- | ----- |
| Glottal entry area | --- | --- | --- | 4.7 | 14.4 | +206% | ----- | ----- | ----- |
| Subglottal area | --- | --- | --- | 255 | 433 | +70% | ----- | ----- | ----- |

None of the data sets provided all the measurements to fully quantify the airway geometry. The CT scans provided airway dimensions above the larynx, the MRI data gave epilaryngeal and pharyngeal airway cross-sectional areas, and the model provided detail for the glottis. The important result is that airway expansions occurred everywhere because the net pressure was positive throughout the airway, even while acoustic pressures riding on top of the steady pressures varied from positive to negative in all regions. However, there was considerable variation across data sets. The largest variation came from measurement of the epilaryngeal exit area and the pharyngeal entry area. The percent change in epilaryngeal exit ranged from a low of 7% in Guzman's data to a high of 177% in the computer model. The percent change in pharyngeal entry area ranged from a low of 32% in the Guzman data to a high of 107% in the single male subject. These cross-sectional area measurements were very difficult to obtain accurately from published CT scans because both the angular orientation and the non-uniform length of the epilaryngeal airway (longer anteriorly than posteriorly, as described in the Methods) could not easily be resolved from the mid-sagittal, coronal, and transverse planes.

Not included in the comparison table are post-SOVT data from Guzman et al. because the computational model has no muscle memory or long-term hysteresis. In Guzman's data, velopharyngeal tract length and epilarynx tract length returned to the pre-SOVT values (47 mm and 14 mm, respectively), pharyngeal entry area remained slightly expanded (50 mm$^2$ instead of 47 mm$^2$ pre-SOVT), and epilarynx exit area remained expanded at 15 mm$^2$.

The biomechanics of the simulation model were not sufficiently advanced to compute vertical displacements of structures with muscle activations. In particular, the lowering of the larynx with intraoral pressure was not calculable with any mechanical laws. Even without larynx lowering, the lateral expansions in the glottal and tracheal regions occurred in the model, as shown in the bottom four rows in Table 1.

It would be tempting to conduct a formal statistical analysis for reliability, accuracy, and significance of the airway expansions listed in Table 1. However, the methodologies were so different (2D versus 3D images, single subject versus multiple subjects, trained versus untrained vocalists, and different lip occlusion resistance) that little can be concluded statistically than what is obvious from the table. In all measurements, airway expansions occurred. It will require much more systematic imaging studies to determine the relative expansions of various sections along the airway. Regarding the modeling results, there is a strong dependence on the assumed stiffness of the airway wall tissues. No variation of wall stiffness (Young's modulus) was programmed from the trachea to the lips. Unfortunately, differential data of this modulus along the airway are scarce and would likely be age- and gender-dependent. Hence, the results in Table 1 should be interpreted as a general order-of-magnitude airway expansions that are likely to have many variations from the trachea to the lips.

## Aerodynamic flow resistances

Flow resistances for steady flow in a pipe or tube stem from air viscosity, vorticity, turbulence, and flow separation from the walls. The so-called kinetic losses (flow separation and vorticity) occur primarily in sudden contractions and expansions of the air channel. Fig 6 show measured resistances for mean airflow through circular cross-section tubes (solid straight lines) and vocal folds (data points) for breathy, normal, and pressed phonation [51]. Both males and females produced these phonations for three conditions of vocal loudness (soft, normal, and loud). The data are averages over eight subjects.

The most interesting result from the human glottal flow resistances is that normal and "pressed" phonation had a small range of airflow (between 0.2–0.3 L/s), but flow resistance rose sharply with loudness. In other words, loudness was changed mainly with more adduction

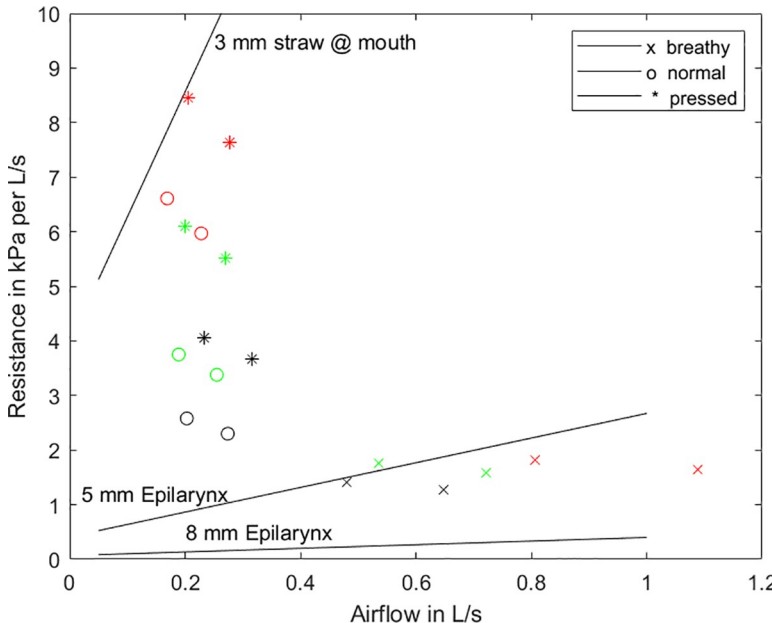

**Fig 6. Steady flow resistances for the glottis at 3 loudness levels (soft—black, medium—green, loud—red) and 3 phonation types (data points), the epilaryngeal airway (lower lines), and the flow resistant tube (upper line).**

of the vocal folds, which created higher flow resistances. To the contrary, breathy voice maintained a low resistance in the glottis, but loudness increased with more flow. For the tubular structures, the flow resistances calculated with Eq (1) in the Methods section had a component that changed linearly with flow. The highest resistance was for an 8 cm long lip tube, 3 mm in diameter. The lowest resistance was for a 2.5 cm long epilaryngeal airway, 8 mm in diameter. This resistance more than quadrupled with a 5 mm diameter. It is clear that the glottal resistances are bracketed by the low-resistance 8 mm epilaryngeal airway and the high-resistance lip tube. It will be argued in the Discussion section that these resistances can be adjusted for maximum aerodynamic power transfer, which then leads also to optimal acoustic source-airway interaction.

## Inertagrams for semi-occluded and open vocal tracts

An inertagram is a plot of the acoustic inertance offered by any section of airway over a range of frequencies. For any given frequency, the acoustic impedance is first calculated as the pressure/flow ratio. It is generally a complex number. The imaginary component is known as the acoustic reactance. If this reactance is positive (inertive), division by the angular (radian) frequency yields the inertance. In a previous paper [52], inertagrams were calculated for a variety of airway shapes with oral semi-occlusions, but the direct comparison between oral semi-occlusion and oral opening was not made. It was shown, however, that large inertance in the 3–4 kHz range was categorically the result of a narrow epilaryngeal airway, regardless of the shape of the remaining airway. Fig 7 shows four shapes of interest for this study, two with pharyngeal widening, as in the vowel /i/, and two with pharyngeal narrowing, as in the vowel /ɑ/. The area functions followed the MRI shapes published by Story et al. [33]. All four shapes have a narrow epilaryngeal airway (5 mm diameter, corresponding to a cross-sectional area of 0.2 cm$^2$, with a 2.5 cm length).

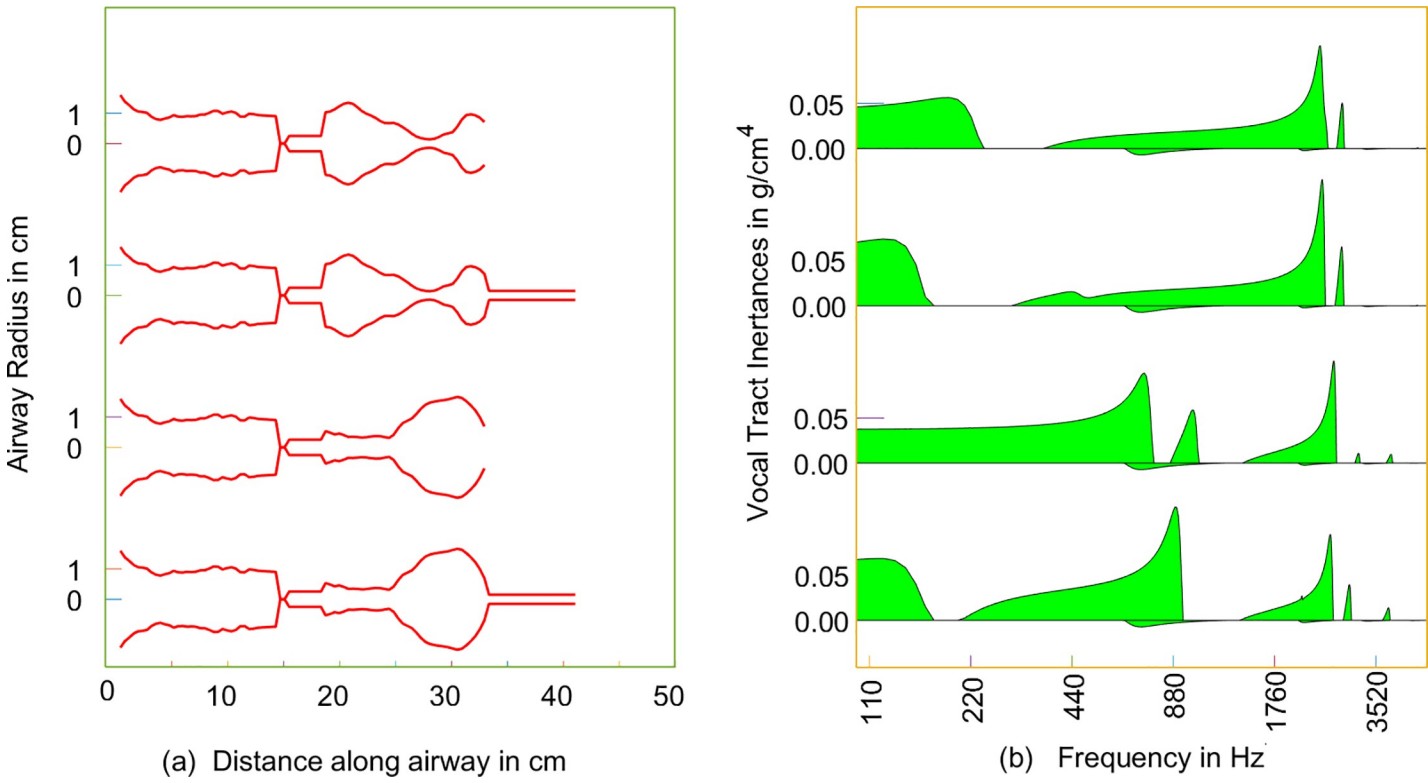

**Fig 7.** (a) Airway shapes for the /i/ vowel (top two) and the /ɑ/ vowel (bottom two) with and without oral semi-occlusions, (b) corresponding inertagrams.

Both pharyngeal shapes were examined with and without a flow-resistant tube at the mouth. The flow-resistant tube had a 3.0 mm diameter and an 8.0 cm length, the same dimensions as in Fig 6. The subglottal configuration, constant for all cases, was also taken from MRI data reported by Story et al., [33] for a male subject.

For the wide pharynx cases, the primary effect of the oral tube was to lower the frequency of first acoustic resonance of the supraglottal airway ($F_1$), which is defined by the first sudden drop in inertance from a high value to zero. $F_1$ is lowered from 250 Hz to 150 Hz with the lip tube. With and without the tube, the inertance increases smoothly from about 300 Hz to above 3000 Hz, where a strong peak occurs that reflects the inertance near resonance of the epilaryngeal airway.

For the narrow pharynx cases, the primary effect of the oral tube was a much larger drop in $F_1$, from around 800 Hz to 200 Hz. There is an increase in peak inertance around 880 Hz, where a clustering of $F_1$ and $F_2$ occurs, as for an open /ɑ/ vowel. Little change is seen in the high-frequency region because the epilaryngeal airway has been kept the same. The subglottal airway produced a small negative (compliant) acoustic reactance around 700 Hz, as seen as a small "tear drop" below the zero line in all inertagrams.

The continuous smooth growth in inertance with increasing frequency in the upper two inertagrams (pharyngeal expansion) is important. Harmonics in this positive inertance territory gain in amplitude [45], which will be demonstrated below with simulation. A continuous positive slope in the growth of inertance with frequency can counteract the typical decay in harmonic amplitude produced by the source. In this way, high-frequency energy can be gained by source-airway interaction instead of excessive vocal fold collision. This is now further demonstrated with simulation.

## Simulation of source-airway interaction with a controlled epilaryngeal airway

In the computational model, the most dramatic airway expansion in the model was the cross-sectional area of the epilaryngeal airway at exit, increasing by 177%. As shown in the inertance plots above, this is precisely the area to keep narrow for impedance matching and high-frequency sound generation. In lieu of shrinking the epilaryngeal airway with muscle control in the simulation model, the wall was stiffened by increasing the Young's modulus to a large number (100 kPa, more than 10 times the original 9.62 kPa value). This simulated the action of the aryepiglottic and transverse interarytenoid muscles to maintain a narrow epilaryngeal airway. Fig 8 shows the result. A semi-occlusion was again imposed at the mouth with a 3 mm flow-resistant tube (starting at distance 35 cm along airway). Note that all regions except the 16–18.5 cm epilaryngeal airway were still expanding. The vertical line near 16 cm in the bottom graph indicates glottal exit. In comparison to Fig 4 (lower graph), glottal expansion was reduced at exit.

With this control, a profound difference was seen in the simulated waveforms of glottal area, glottal airflow, and oral acoustic pressures. Figs 9 and 10 show time waveforms (left) and their corresponding frequency spectra (right) with and without epilaryngeal airway expansion, respectively. Note the lack of high-frequency harmonic energy when expansion is allowed (Fig 9). Only the first three harmonics are visible in the oral pressure signal.

In contrast, when the epilaryngeal airway is not allowed to expand, a high-frequency ripple can be seen on the oral pressure waveform (Fig 10). The spectrum of the oral pressure waveform shows high-frequency components in the 3–4 kHz region.

The contrast between an expanded versus a controlled epilarynx area was seen in the Guzman et al. [31] data and the Guzman et al. [53] data. The 2013 study [31] was on a trained subject who had apparently mastered epilaryngeal airway control. Expansion was inhibited. To the contrary, the untrained subjects in the 2017 study [53] did allow expansion. They had

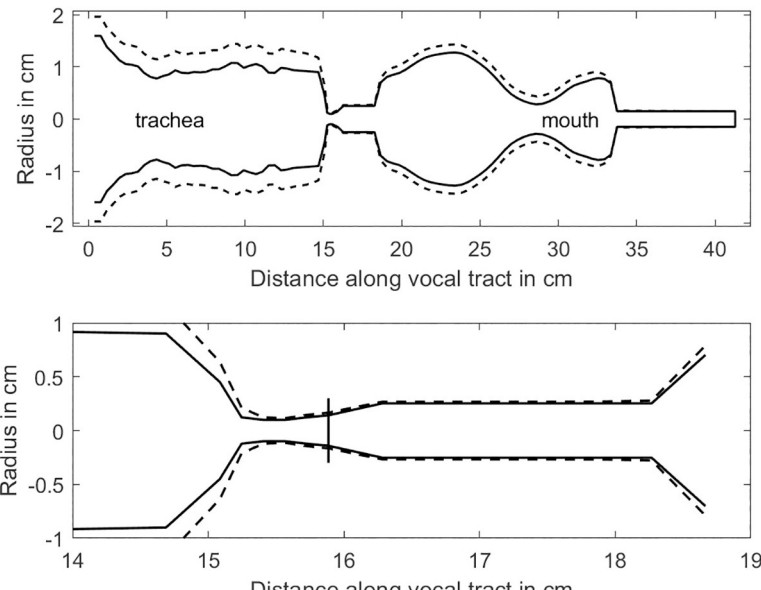

**Fig 8. Expansion of the airway in a simulation model with a 3.0 mm diameter oral semi-occlusion and a lung pressure of 2.5 kPa.** The stiffness of the epilaryngeal airway wall was increased to prevent expansion. (Top) entire airway, and (bottom) glottal and epilaryngeal sections.

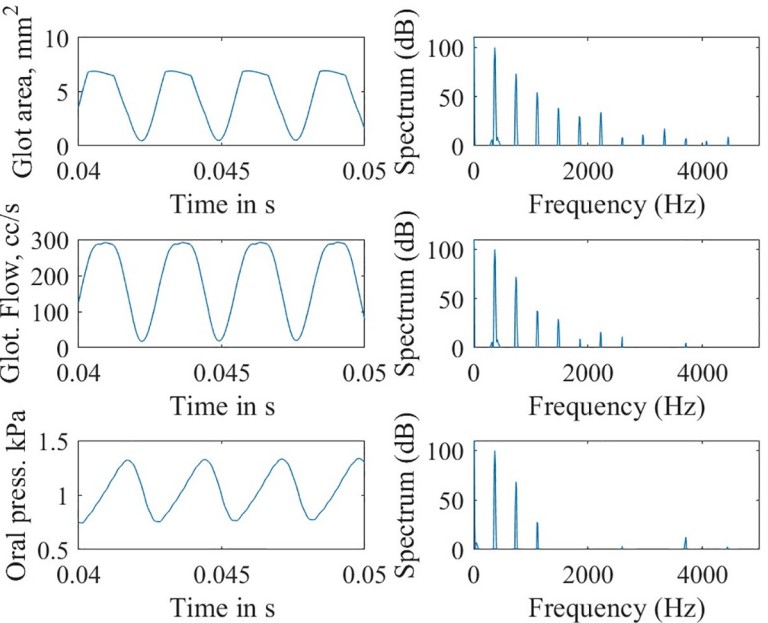

**Fig 9. Waveforms and spectra for the simulation with epilaryngeal airway allowed to expand.**

never been exposed to semi-occluded vocal tract exercises and had no training in producing variable voice qualities.

## Discussion

The results of this study show that optimal interaction between the sound source and the airway can be facilitated with an oral semi-occlusion in the airway. With a narrow flow-resistant

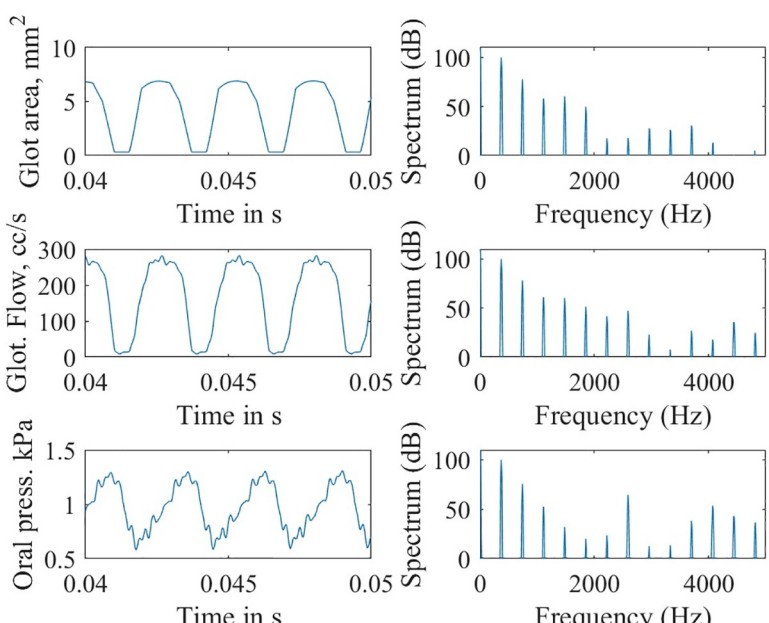

**Fig 10. Waveforms and spectra for the simulation with epilaryngeal airway not allowed to expand.**

tube between the lips (on the order of 3 mm in diameter), a general expansion of all airways between the lungs and the semi-occlusion is produced passively. This expansion then provides more opportunity for wide-narrow differentiation in airway geometry. Wide-narrow differentiation appears to be advantageous for phonetic and voice quality contrast, especially when source-airway interaction exists. On its own, the global airway expansion would lower the flow resistance and the acoustic impedance of the airway, making it much smaller than the glottal source impedance (except at a resonance frequency of the vocal tract). This low airway impedance is advantageous for source-airway independence, but it is not advantageous for maximum power transfer from the source to the airway according to conventional transmission-line theory [54]. Maximum power transfer and maximum source-airway interaction occur when the impedance of the source matches the impedance of the transmission system carrying the power to the load.

What is the source impedance and what is the impedance of the transmission system that carries power to the load in vocalization? For the steady airflow component, the impedances are flow resistances. The flow resistance of a narrow tube at the mouth is large enough that it can be considered nearly the total airway load resistance ($R_L$), but the epilaryngeal resistance ($R_e$) adds a small component. The combined resistance of the glottis ($R_g$) and the tracheal resistance ($R_t$) can be considered the source resistance for transfer of power from the larynx to the airway above the larynx. The steady flow data showed that the source resistance can approximate the supraglottal airway resistance when an oral semi-occlusion is imposed. However, when the mouth is opened, an equalization between source resistance, $R_g$ and epilaryngeal or pharyngeal resistance is needed for maximum power transfer. Fortunately, source resistance and epilaryngeal resistance tend to change in opposite directions. Our simulation data showed that $R_g$ decreases when $R_e$ increases. As the epilaryngeal airway resistance is increased with active narrowing, the glottal resistance is reduced by the increase in passive pressure from the supraglottal airway, spreading the vocal folds apart and thereby reducing $R_g$. Based on maximum power transfer from the source to the vocal tract, it is believed (but not yet proven) that the system self-organizes to balance the two resistances. Active control with laryngeal and pharyngeal muscles is likely involved. The semi-occlusion at the mouth becomes the prosthetic device to facilitate muscle motor memory for maintaining a lower glottal resistance. An increased epilaryngeal or pharyngeal airway resistance can then match the glottal resistance for maximum aerodynamic power transfer.

$$(\mathbf{R_g} \sim \mathbf{R_e})$$

With regard to acoustic energy transfer to the airway, the inertance of the vocal tract air column is raised by epilaryngeal narrowing. With this increased inertance, the supraglottal acoustic pressures assist the vocal folds in vibration and the glottal flow increases in harmonic content. The typical decay of harmonic amplitude with increased frequency can be offset by a positive slope in airway inertance with frequency. It has been shown that an optimal ground configuration of the airway is the combination of an epilaryngeal contraction and a pharyngeal expansion. This combination produces the best overall positive inertance slope in the 300–4000 Hz range. As a minor contrast, the combination of a narrow epilarynx tube and a narrow pharynx is effective for increasing frequency components in the 200–2000 Hz range, but with more variable enhancement above 2000 Hz due to the inertance gaps between the higher resonances.

This study did not address the specific active muscular control needed for variations in epilaryngeal and pharyngeal spaces. The oblique interarytenoid and the aryepiglottal musculature is difficult to probe with electromyography. We showed, however, that preventing the

expansion of the epilaryngeal airway by stiffening the wall was a substitute strategy for maintaining the narrow configuration. These adjustments may vary significantly with the desired sound quality and sound intensity desired by the vocalist. A range of 0.2–0.5 cm$^2$ for an average epilaryngeal tube diameter produces significant spectral changes [47]. Driven by acoustic contrasts, various species may have developed unique abilities to isolate specific laryngo-pharyngeal muscles for variation in airway geometry, perhaps counteracting the reflexive gestures that evolved for airway protection

## Methods

Multiple experimental and theoretical approaches were combined here to address the hypotheses, including computer simulation, retrieval of linear and area measures from CT and MRI imaging, impedance analysis, and acoustic spectrography. The data of Guzman et al. [53] were used to extract airway geometry relevant to this study. Guzman et al. published CT scans of 10 subjects (five males and five females) with no vocal training and mild hyper-functional vocal disorders. These authors showed that critical dimensions of the upper airway were widened passively as a result of a steady pressure built up behind an oral semi-occlusion. These data were augmented with data from a classical MRI study on a single male subject [33] to get a few more details in the epilaryngeal region. Aerodynamic and acoustic impedance calculations were performed on models of the airway for a variety of configurations, with and without semi-occlusions. The entire system (subglottal, glottal, and supraglottal airways) was then energized computationally with a lung pressure to simulate airflow, self-sustained vocal fold oscillation, and wave propagation. Frequency spectra were computed from the time waveforms obtained from the simulation.

### Extraction of critical airway dimensions from CT and MR imaging

The subjects of Guzman et al. [53] produced a sustained [a:], followed by sustained phonation into a straw, and then repeated the sustained [a:]. After 15 minutes of vocal rest, the same procedure was performed with a stirring straw. Several anatomic distances and area measures reported by Guzman et al. from the CT images were utilizable in our computational model, but additional cross-sectional measures needed to be obtained from the CT images. The most difficult measurements were the effective length and the exit area of the epilaryngeal airway due to its tilt relative to the remainder of the airway. The epilaryngeal airway is partially a tube within a tube. The tube length, about 2–3 cm long, has an angular orientation with respect to the pharynx and the piriform sinuses. The exit is bounded ventrally by the epiglottis and dorsally by the arytenoid cartilage, the piriform sinuses, and the pharyngeal wall. Acoustically, the laryngo-pharynx is essentially a narrow tube surrounded by a wider tube, formed by the thyroid cartilage and the posterior wall [55]. We showed in the Results that the narrow epilaryngeal airway plays a major role in impedance matching between the source and the remainder of the airway.

Some additional single-subject data were obtained from a classical MRI study by Story et al. [33]. This study was extensive in terms of vocal tract shapes. The male subject produced ten vowels, two semi-vowels, and six consonants. Each of the 18 shapes was quantified with 44 cross-sectional areas perpendicular to the airway axis. The length of each section was 0.3968 cm.

While this study did not include a flow-resistant straw or tube as an oral semi-occlusion, it included three nasal consonants, which are considered airway semi-occlusions. We compared the epilaryngeal airway dimensions of the vowel /ɑ/ to that of the consonant /m/. From the combined CT and MRI images, we obtained airway data as listed in Table 1.

## Computational simulation

As briefly described in the Results section, a tubular airway model was constructed mathematically with variable diameters as shown in Fig 3. It included, from left to right, a trachea, a glottis (five short narrow sections that could sustain self-oscillation), an epilaryngeal airway, a supraglottal tract, and a flow-resistant tube attached at the mouth (long narrow section) as a semi-occlusion. The epilaryngeal airway included the ventricle, the ventricular fold glottis, and the vestibule [47]. The model also included a nasal tract (shaded area above the zero-diameter line), but the velum was kept closed for the computations because velar closure is usually part of oral semi-occlusion exercises. Fluid flow and acoustic wave propagation in all airway sections (including the source sections, to be described below) were calculated on the basis of conservation of momentum and mass transfer using the Navier-Stokes and continuity equations for non-steady, compressible airflow [46]. Fluid pressures on the surfaces of the sections provided the driving forces for all wall expansions and vibrations. For a divergent glottis, for which flow separation occurs with a 20% or greater expansion [56,57], the pressure against the glottal wall was assumed to be the supraglottal vocal tract input pressure. Radiation from the distal end of the flow-resistant tube (the semi-occlusion) was computed with a low-frequency lumped-element model [9]. The viscoelastic wall properties for all sections except the vocal fold sections were chosen according to Titze et al. [46]: Young's modulus = 9.62 kPa, shear modulus = 1.67 kPa, mass per unit area = 1.5 $g/cm^2$, and damping ratio = 1.26 (over-damped so that wall vibration was not self-sustained in airway sections that were not considered sound sources).

Details of the sound-source region, a short and narrow series of sections, is shown in Fig 11. A simple self-oscillating vocal fold model, not specific to any species, age, gender, or vocal fold morphology was constructed with 5 short subsections. The sections were defined by serially coupled soft-wall ellipses (each 1.6 mm in length in the caudal-cranial direction), giving the vibrating tissue an overall thickness of 8.0 mm in the caudal-cranial direction. The minor diameters (also known as the pre-phonatory glottal widths) were used to create different medial surface contours caudal to cranial, while the major diameters (also known as vocal fold length, ventral to dorsal) were kept at 10 mm. The elliptical sections allowed for a gradual change from circular geometry in the trachea and supraglottal tract to a slit-like ellipse for glottal geometry. Five sections were used so that the glottis could be convergent, divergent, or bulging (convergent at entry and divergent at exit). Fig 11 shows a convergent-divergent contour produced for medial surface bulging. Other shapes possible were parallel surfaces, linearly converging, and linearly diverging. The viscoelastic properties of the oscillating wall were patterned after the two-mass model of Ishizaka and Flanagan [20], with a Young's modulus = 4.0 kPa, a shear modulus = 1.0 kPa, mass per unit area = 0.3 $g/cm^2$, and damping ratios of 0.2 on the lower three (left) sections and 0.6 on the upper two sections (right). This produced a natural tissue frequency of 130 Hz in each section. Higher frequencies were obtained by raising the Young's modulus and lowering the mass per unit area. With this model, the geometry of the

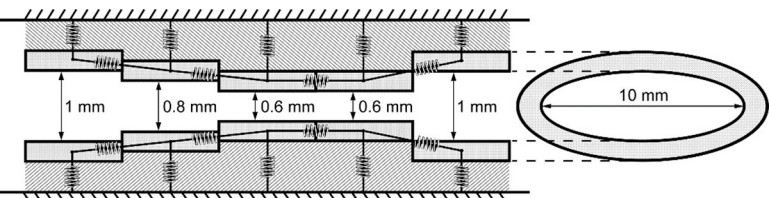

**Fig 11. Five serial elliptical vocal fold sections for self-sustained oscillation.**

semi-occlusion was varied so that its effect on source-airway interaction could be determined. In particular, the effect on the vocal fold surface positioning and general airway expansion were calculated.

## Steady flow resistance of airway segments, including the glottis

Pressure-flow relations in an airway can be divided into two parts, those related to uni-directional respiratory flow, and those related to oscillatory flow superimposed on the respiratory flow as a wave. The pressure/flow ratios at the entry of an airway section are generally quantified as impedances, with a real part known as resistance R, and an imaginary part known as reactance X. However, for the steady respiratory component, the reactance is zero and the pressure-flow relations are quantified with flow resistances.

Smith and Titze [58] conducted extensive pressure-flow measurements on tubes and straws of various lengths and diameters. All were in the range of airway dimensions and typical tube or straw extensions, 3–24 cm in length and 1.8–9.7 mm in diameter. An empirical equation was published for the entire set of 32 geometries:

$$R = \left(3.7631 \times 10^{-7} \frac{L}{D^{4.4997}} + 1.0268 \times 10^{-6} \frac{1}{D^{4.0416}}\right) Q + \left(3.9913 \times 10^{-9} \frac{L}{D^{5.0089}} + 8.0169 \times 10^{-7} \frac{1}{D^{3.7696}}\right) \quad (1)$$

In the above, L is the tube length in m, D is the tube diameter in m, and Q is the airflow in L/s. This equation was used to calculate the flow resistance of the epilaryngeal airway for several diameters, as well as the flow resistance of tubes used as an oral semi-occlusion. The flow resistance of the human glottis (airspace between the vocal folds) was available from Konnai et al. [51]. They conducted glottal pressure-flow measurements on three males and five female human subjects for three levels of loudness (soft, medium, and loud) and three conditions of vocal fold adduction (breathy, normal, and pressed). Eighteen data points were available for comparison to the bench measurements conducted by Smith and Titze [58].

## Acoustic impedance calculations and generation of inertagrams

Some key acoustic pressure-flow relations of the airway (with or without a tube extension) were described in terms of the input impedance Z,

$$Z = R + i\,X \quad (2)$$

The real part R is known as the acoustic resistance of the airway and the imaginary part X is known as the acoustic reactance of the airway. A non-zero reactance guarantees that there is a phase difference between the pressure and flow.

The resistance is generally a positive number. It quantifies the energy losses in the airway. The reactance, which is frequency-dependent, can be positive or negative. Positive reactance, also known as inertive reactance, causes a phase delay in the time-varying airflow relative to the applied pressure. Negative reactance, also known as compliant reactance, causes a phase advancement of the airflow relative to the applied pressure. The words inertance and compliance were carefully chosen by engineers and physicists in multiple disciplines because inertia implies a delay in response and compliance implies an advanced or anticipated response to the stimulus. Positive (inertive) reactance X can further be written as

$$X = 2\pi f\,I, \quad (3)$$

where I is defined as the inertance and f is the frequency of interest. A plot of the inertance over a range of frequencies has been labeled an inertagram, as shown in Fig 7B. The greater the inertance at any frequency, the greater the excitation of that frequency at the sound source

[45]. Vocal tract inertance can also lower the phonation threshold pressure [15,22], making it easier for the vocal folds to be set into vibration.

The methodology of computation follows the approach outlined by Story et al. [44]. For the supraglottal airway, it begins with the radiation impedance at the distal end of the airway. This radiation impedance is already in the form of an inertance in parallel with a radiation resistance. Thus, an initial pressure/flow ratio already exists as an airway termination. This ratio is then transmitted through a series of tube sections of given lengths and given diameters with a 2x2 matrix calculation. There is no limit to the number of sections that can be concatenated to form a complex airway configuration. All computations are in the form of complex numbers.

In Fig 7B, the shaded (green) areas in the inertagrams indicate the magnitude of inertance at any frequency. It was pointed out that the narrow epilaryngeal airway above the vocal folds dramatically raised the overall inertance, whereas the pharyngeal airway altered the distribution of the inertance in the higher frequency regions. For the subglottal airway, the calculation begins with an impedance estimate at the tracheal bifurcation [59]. Again, the pressure-flow ratio is transmitted through the sections of the trachea to compute the final ratio below the glottis. Owing to a 180-degree phase shift between flow into the glottis for the subglottal system versus flow out of the glottis for the supraglottal system, a compliant reactance is converted into an equivalent inertive reactance and then into an inertance [45]. This equivalent inertance is entered into the inertagram as an extension below the zero line.

## Conclusions

This study has offered a physical and physiological understanding of the efficacy of vocalization with an oral semi-occlusion. Practical messages from this study are that the oral pressure resulting from a highly flow-resistant semi-occlusion widens the entire airway passively, providing more opportunity for airway shape contrasts. If the epilaryngeal airway is selectively prevented from the passive expansion with muscular control, source-airway interaction can be optimized with impedance matching, both aerodynamically and acoustically. Vocalization with this high-resistance load at the mouth promotes an increase in flow resistance in the larynx, but with narrowing the epilaryngeal airway above the vocal folds rather than pressing the vocal folds together for increased glottal resistance. The process of source resistance balancing has likely been explored by multiple species that vocalize with small and large airway openings.

Perhaps the most important finding is that training with oral semi-occlusions can carry over into vocalization with an open mouth if optimal epilaryngeal-pharyngeal configurations are maintained with some sensory or motor memory. The sensation of lower glottal resistance with slightly separated vocal fold surfaces, together with the sensation of more efficient sound production with impedance matching and lower phonation threshold pressure, may be retained. This may lead to the avoidance of "pressed voice", a vocalization associated with increased vocal fold contact pressure and related tissue trauma.

Use of full lung pressure with semi-occlusions seems to present no problem because greater lung pressure does not produce more vocal fold collision. In this study, a 2.5 kPa lung pressure, three times greater than a typical pressure for vowel production with an open mouth, produced only a small closed phase in the glottal area and glottal flow. The spreading of the vocal folds is proportional to the internal pressures produced, thereby negating the effect of excessive collision experienced with open-mouth phonation at high lung pressures. Wide pitch ranges can then be explored with semi-occluded vocal tract exercises without injury from excessive collision trauma.

Not yet fully explained with quantitative biomechanical modeling is the effect that a balance between subglottal and supraglottal pressures may provide an optimal vertical position of the

vocal folds. CT imaging shows that the larynx lowers with a semi-occlusion, therewith lengthening the epilaryngeal airway. The subglottal area may effectively widen with a lowered larynx due to a sharper transition from the trachea to the entry of the glottis. Both of these airway adjustments are predicted to optimize acoustic power and greater spectral content [52].

## Author Contributions

**Conceptualization:** Ingo R. Titze.

**Data curation:** Ingo R. Titze, Anil Palaparthi, Karin Cox.

**Formal analysis:** Ingo R. Titze, Anil Palaparthi.

**Funding acquisition:** Ingo R. Titze.

**Investigation:** Ingo R. Titze, Anil Palaparthi, Karin Cox, Amanda Stark, Lynn Maxfield, Brian Manternach.

**Methodology:** Amanda Stark, Lynn Maxfield.

**Project administration:** Lynn Maxfield.

**Software:** Ingo R. Titze, Anil Palaparthi.

**Validation:** Ingo R. Titze, Anil Palaparthi.

**Visualization:** Ingo R. Titze, Anil Palaparthi.

**Writing – original draft:** Ingo R. Titze, Anil Palaparthi, Karin Cox, Amanda Stark, Lynn Maxfield, Brian Manternach.

**Writing – review & editing:** Karin Cox, Amanda Stark, Lynn Maxfield, Brian Manternach.

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
