## [Decision Letter · Decision Letter 0]

1 Dec 2020

Dear Dr. Titze,

Thank you very much for submitting your manuscript "VOCALIZATION WITH SEMI-OCCLUDED AIRWAYS IS FAVORABLE FOR OPTIMIZING SOUND PRODUCTION" for consideration at PLOS Computational Biology. As with all papers reviewed by the journal, your manuscript was reviewed by members of the editorial board and by several independent reviewers. The reviewers appreciated the attention to an important topic. Based on the reviews, we are likely to accept this manuscript for publication, providing that you modify the manuscript according to the review recommendations.

Sincerely,

Alison Marsden

Associate Editor

PLOS Computational Biology

Natalia Komarova

Deputy Editor

PLOS Computational Biology

[LINK]

Reviewer's Responses to Questions

**Comments to the Authors:**

Reviewer #1: VOCALIZATION WITH SEMI-OCCLUDED AIRWAYS IS FAVORABLE FOR OPTIMIZING SOUND PRODUCTION

Abstract: This sentence is not clear: “It is hypothesized that vocalization with a semi-occluded airway has an overall widening effect on the airway” and suggests that the act of semi-occluding is to widen the airway (I am assuming that is not what is meant)

L. 58: Plural needed – ‘received by listeners’ (unless there is a reason that I have not spotted to be singular?

L. 61: The comparison with sounds radiated from the skin seems out of place here – or I am missing the point?

L62: This sentence is also a bit of a non-sequitur: narrow the tract, sound is more efficient (OK) then you can open the mouth (after practice). This implies (to me at least) that practice can obviate any advantage of a narrowed tract.

L. 64: Here there is a jump from the ‘narrow tube at the mouth’ to the ‘airspace directly above the vocal folds’ – another non-sequitur to me. How far up the tract is ‘directly above’ the vocal folds – the last sentence indicates that this allows a large variety of sounds; I assume not all the vowels etc.?

L. 88: Is there evidence for ‘optimally’ to support the statement after ‘presumably’? I do not see that the semi-occluded tract exercises proves that this statement is supported: ‘optimal airway configuration and optimal posturing and shaping of the vocal folds for vibration’.

L. 92: For no brass players a reference for ‘the variable interaction experienced by brass instrument players who mute the instrument at the bell’.

L. 103: The word ‘sharp’ here is undefined and could be a hostage to fortune – isn’t ‘square or saw-tooth-like’ sufficient; indeed, might ‘pulse’ be more appropriate than ‘square’?

L. 106: I think something should be added to exemplify what the ‘acoustic interaction between ..source increases’ means in practice – this paper will be of great interest to singers.

L. 110: What is the relationship between impedance and source vibration frequency and how might it affect the system?

L. 109: Is there any condition on which ‘portion’ of the airway might be narrowed in terms of the change in impedance?

L. 119: Just to be clear, state what sound was phonated into a drinking straw in Laukkanen’s experiment.

L. 133: A bit of a non-sequitur here – on the one hand humans do not use widening much but when we need to we can widen the tract (deep breathing is an interesting case given the breath control care that singers employ so I wonder if this leads to more tract widening strategies arising from more conscious breath control?).

L. 169: Justify the decision to use 107 short circular sections and why an 8 cm tube at the lips.

Figure 3: It would be useful to have a few landmarks indicated on the diagram to aid readers’ geography such as the trachea, glottis, velum, lips, nasal tract.

L. 169 and L. 179 differ in the length quoted for the flow resistant tube (8 cm and 10 cm respectively) – cannot both be right!

L182. I would say that 300 Hz is in the middle of a typical singing range - should be made clear it is not for speech range.

L182. Brackets do not close.

L183. Reference or show the obtaining of this frequency.

L.185. Define clearly the meaning of ‘post-pressure’.

P.190. Make clear exactly where the nearly doubling is located – it is not clear to me by eye on the figure. Is it the fact that it is near doubling that is crucial or just the fact that there is a significant expansion?

Figure 5: I note that these effects are plotted for 2.5 kPa lung pressure – how do they vary across typical lung pressure ranges (if that can be easily added) – this would place this plot in a useful place for practitioners and at the very least would benefit from a comment I think.

L. 233: Remove ‘that’.

Table 1: What are the error margins in these measurements? Line 238 indicate that the cross-sectional area measurements were “very difficult to obtain accurately” without quantifying the difficulty.

Figure 6: I find the plotted points confusing – there are 6 plotted points for each of three phonation types that are measured from 8 subjects (males and females – 4 each I assume) from whom the data are averaged phonating at three loudness conditions. So are the 6 points 3 loudness levels from males and females – I think reading loudness from the plot should be very clear in the figure caption so the points can be unambiguously identified (unless I have not understood it).

L. 292: Noted that the sub glottal measurements were taken from Story et al, but it is not clear that the modelled measurements were appropriate tract dimensional matches (which I assume they were).

Fig. 7 – a lovely result.

Fig. 8: Make it clear where the 3 mm diameter semi occlusion is in the figure (I assume it is the cylindrical part from about 16.5 mm to 18.5 mm on the lower plot) and also what the vertical line is indicating on the lower plot.

L. 374: ‘e’ missing on ‘the’.

L. 380: Is there a source to back up the statement ‘It is believed …’?

L. 399: Can the ‘variable enhancement above 2000 Hz’ comment be supported?

L. 404: As a practical conclusion for singers, could this be dimensioned in terms of what sort of adjustment extents would be appropriate to cover different sound qualities and intensities?

L. 571: I am not sure what the message is here – are you advocating the avoidance of pressed voice or of pressed voice with reduced VF contact or suggesting the motor sensory memory needs to ‘learn’ to avoid it?

Reviewer #2: The manuscript provides a strong and thorough outline with detailed results. However, a few minor revisions would aid in the overall flow the manuscript, especially with respect to the discussion section. First, the methods section should be moved before the discussion section so that the reader understands how the results which are being discussed we obtained. Second, a statistical analysis, or argument as to why in this situation it is not needed, would be useful to create a more compelling argument with the data, especially for Table 1. Given the discrepancy in data between the multiple studies cited and the novel data provided, an analysis would be useful to show whether these results agree with each other, and to provide more robustness behind the novel Young's modulus based simulation used. If the author' feel that this is not needed for what is being argued, a more explicit reason should be provided to explain the variation in the expansion of the vocal tract and whether this variation is reasonable given the various experiments. This could be done with a single sentence or two, which is why it is felt that only minor revisions are needed.

Reviewer #3: General comments

Thank you so much for allowing me to review this very interesting manuscript. I think it is definitely a nice contribution to the field and adds new content to a relevant topic in voice training and therapy: the semioccluded vocal tract exercises. The paper is appreciated for its attention to many details in research design and study execution. Some minor additions I think could contribute to the paper.

Introduction

Could authors provide to readers a brief explanation about why aerodynamic and acoustic interaction between the source and the airway play a major role when the sound source is located deep within the airway? Why is it more efficient?

Line 82: is there evidence that musical wind instruments exhibit the same interaction phenomena? I mean, is there evidence for instance that lips vibrate in an easier way because of the long trumpet tube? If so, could authors provide some references?

Line 87: what do you mean by optimal sound production? More efficient in terms of energy conversion? More economic? Please briefly explain this.

Line 103-105: Could you please mention (in parentheses) that you are talking about acoustic-aerodynamic interaction and mechano-acoustic interaction respectively?

Line 106-107: Some references are needed to support that statement.

Line 112-113: Statement about ear channel needs some references.

Results and discussion

Line 180: Since in clinic lung pressure is usually measure in cm of H2O, could authors provide the amount of H2O equivalent to 2.5 kPa?

Why only a 3 mm tube was used for airway expansion? Please provide a brief explanation.

Line 206-208: “Note that the wider oral tubes (5 mm and 6 mm) produced only a modest glottal expansion, whereas the 3 mm tube dramatically separated the vocal folds at glottal entry and exit” Is this caused by the greater oral pressure generated by thinner tubes? If so, provide a brief explanation to readers.

Line 348: Tube used in Guzman et al. 2013 and Guzman et al. 2017 differ in inner diameter. I think, Guzman et al. 2013 used narrow straw and Guzman et al. 2017 used regular drinking straw (wider inner diameter). Could this be an additional explanation for the epilarygeal tube configuration (narrower) reported during tube phonation in Guzman et al. 2013? Maybe the epilarygeal tube narrowing reported by Guzman et al. 2013 was a reaction of the source system, specifically of the epilaryngeal airway resistance (Re) (active narrowing) to match the resistance at the lips (narrow straw) in order to increase the maximum power transfer. Since the resistance at the lips was lower (wider straw) in Guzman et al. 2017, maybe the source system, specially Re did not react in the same way (no active narrowing) because there was no need to get narrower to increase the power transfer. What do you think about this? It would be interesting to include this at this part of the manuscript or in the discussion section.

Furthermore, in a laryngoscopic study by Guzman et al. 2013, it was also found that the degree of airflow resistance at the lips affect the degree of anterior-posterior laryngeal compression (i.e. epilaryngeal tube narrowing). Phonation into a thin straw and tube submerged 10 cm into the water produced the greatest degree of epilaryngeal tube narrowing compared to a wider straw and other SOVTE.

Peltokoski et al, 2015 found similar results in a laryngoscopic study. They reported a narrower epilaryngeal tube during phonation in water (high resistance) compared to tube phonation in air (lower resistance). I think this could be a nice discussion. What do authors think?

Line 377-380: “As the epilaryngeal airway resistance is increased with active narrowing, the glottal resistance is reduced by the increase in passive pressure from the supraglottal airway, spreading the vocal folds apart and thereby reducing Rg.”

Regarding this paragraph, I assume that the model implemented in this study did not include a possible compensatory muscle and aerodynamic behavior from the subglottis. It has been previously observed that when there is an increment in oral pressure caused by high airflow resistance voice exercises, there is an automatic compensation, an increment of the subglottic pressure (e.g. Radolf et al. 2014, Guzman et al. 2015), which in turn, tend to increase vocal folds adduction in some cases. If the increment of subglottic pressure is greater than the increment of oral pressure, then the transglottal pressure will increase. On the other hand, if the oral pressure increment is greater than the subglottic pressure, the transglottal pressure will decrease. Therefore, it could be the case that in human subjects the increment of epilaryngeal airway resistance with active narrowing, not necessarily produces a reduced glottal adduction by spreading the vocal folds apart. It could also happen that this compensatory behavior (increased Psub as a reaction of the increased Poral) is only an immediate automatic reaction, but with some sensory or motor memory this could be properly train not to increase transglottal pression, as authors stated in the conclusion. I think this entire point could also be a nice aspect to discuss in the paper.

Conclusion

Line 573: “Use of full lung pressure with semi-573 occlusions seems to present no problem because greater lung pressure does not produce more vocal fold collision”.

As mentioned above this could occur only if transglottal pressure does not substantially increase during semiocclusions (because of an increased compensatory Psub). Some studies have reported an increased Ptrans (e.g. Guzman et al. 2015).

Moreover, an increased transglottal pressure and a subsequent increased vocal folds adduction and collision should not be considered always as a detrimental behavior during SOVTE. In voice conditions such as vocal folds paralysis, presbiphonia, or hypophonia caused by Parkinson Disease, an increased vocal folds impact stress is totally desirable. Previous high-speed digital imaging studies have suggested that high airflow resistance SOVTE tend to produce and increase in vocal folds impact stress (Laukkanen et al. 2007, Guzman et al. 2017)

**Have all data underlying the figures and results presented in the manuscript been provided?**

Reviewer #1: Yes

Reviewer #2: None

Reviewer #3: Yes

PLOS authors have the option to publish the peer review history of their article (what does this mean?). If published, this will include your full peer review and any attached files.

Reviewer #1: No

Reviewer #2: No

Reviewer #3: No
---

## [Decision Letter · Decision Letter 1]

26 Jan 2021

Dear Dr. Titze,

We are pleased to inform you that your manuscript 'VOCALIZATION WITH SEMI-OCCLUDED AIRWAYS IS FAVORABLE FOR OPTIMIZING SOUND PRODUCTION' has been provisionally accepted for publication in PLOS Computational Biology.

Best regards,

Alison Marsden

Associate Editor

PLOS Computational Biology

Natalia Komarova

Deputy Editor

PLOS Computational Biology

Reviewer's Responses to Questions

**Comments to the Authors:**

Reviewer #1: I believe all my original comments have been satisfactorily dealt with and I have no further comments.

Reviewer #2: After the edits done, I feel that the edits sufficiently respond to the comments of myself and the other reviewers. Most of the comments were concerning sentence clarity and have thus been resolved by edits to word choice and sentence structure. Additionally, the clarifications to methods and processes used in this study were helpful in resolving lingering questions about the study and the results derived from it. Overall, I do not feel any further edits are needed and any specific wordings are, at this point, a matter of personal preference up for reasonable disagreement.

This paper seems to be a very valuable contribution to the field of SOVTEs and I look forward to future work to add to the novel innovations developed in this paper.

Reviewer #3: I recommend the revised version of this manuscript for publication without additional comments

**Have all data underlying the figures and results presented in the manuscript been provided?**

Reviewer #1: Yes

Reviewer #2: Yes

Reviewer #3: Yes

PLOS authors have the option to publish the peer review history of their article (what does this mean?). If published, this will include your full peer review and any attached files.

Reviewer #1: No

Reviewer #2: No

Reviewer #3: No

---

## [Editor Report · Acceptance letter]

8 Mar 2021

PCOMPBIOL-D-20-01702R1 

VOCALIZATION WITH SEMI-OCCLUDED AIRWAYS IS FAVORABLE FOR OPTIMIZING SOUND PRODUCTION

Dear Dr Titze,

I am pleased to inform you that your manuscript has been formally accepted for publication in PLOS Computational Biology. Your manuscript is now with our production department and you will be notified of the publication date in due course.

With kind regards,

Alice Ellingham
